# Principles for Controlling the Shape Recovery and Degradation Behavior of Biodegradable Shape-Memory Polymers in Biomedical Applications

**DOI:** 10.3390/mi12070757

**Published:** 2021-06-27

**Authors:** Junsang Lee, Seung-Kyun Kang

**Affiliations:** 1Department of Materials Science and Engineering, Seoul National University, Seoul 08826, Korea; tpflwkdhk@snu.ac.kr; 2Research Institute of Advanced Materials, Seoul National University, Seoul 08826, Korea; 3Institute of Engineering Research, Seoul National University, Seoul 08826, Korea

**Keywords:** shape memory effect, biodegradable polymer, thermomechanical property, degradation mechanisms, biomedical application

## Abstract

Polymers with the shape memory effect possess tremendous potential for application in diverse fields, including aerospace, textiles, robotics, and biomedicine, because of their mechanical properties (softness and flexibility) and chemical tunability. Biodegradable shape memory polymers (BSMPs) have unique benefits of long-term biocompatibility and formation of zero-waste byproducts as the final degradable products are resorbed or absorbed via metabolism or enzyme digestion processes. In addition to their application toward the prevention of biofilm formation or internal tissue damage caused by permanent implant materials and the subsequent need for secondary surgery, which causes secondary infections and complications, BSMPs have been highlighted for minimally invasive medical applications. The properties of BSMPs, including high tunability, thermomechanical properties, shape memory performance, and degradation rate, can be achieved by controlling the combination and content of the comonomer and crystallinity. In addition, the biodegradable chemistry and kinetics of BSMPs, which can be controlled by combining several biodegradable polymers with different hydrolysis chemistry products, such as anhydrides, esters, and carbonates, strongly affect the hydrolytic activity and erosion property. A wide range of applications including self-expending stents, wound closure, drug release systems, and tissue repair, suggests that the BSMPs can be applied as actuators on the basis of their shape recovery and degradation ability.

## 1. Introduction

Shape memory material (SMM) is a material capable of recovering the original shape in response to particular stimuli. The temperature-triggered shape memory effect (SME) has been observed in metal alloys such as gold–cadmium alloy [1] and nickel–titanium alloy [2] after the mid-20th century. The SME of metal alloys is based on phase transformation via temperature control with mechanical deformation. SMM is widely used in aerospace technologies [3], textile engineering [4], robot technologies [5] and biomedical applications [6,7,8,9,10,11]. For example, space-deployable structures for satellite, antenna, and solar arrays [3] and dynamic textiles, which sense and react to temperature changes, have been used as window treatments, partitions, and wall hangings [4]. SMM is increasingly used as an actuator of self-propelling soft-robotics-based systems to provide mobility by programing multiple shape transitions [5].

Effective manufacturing of shape-memory nickel–titanium alloy (SMA) by using spark-plasma sintering method was reported, increasing the purity of the product by preventing the chemical inhomogeneity and oxidation introduced during the multi-step conventional method [12]. Combining shape-memory alloy with polymer successfully produced multifunctional composites using the shape-transition behaviors of different materials [13] and stability against inflammation by preventing the release of metal content considering interfacial adhesion [14]. However, the high melting temperature and reactivity of SMAs continue to make their manufacture more complex than shape memory polymers (SMPs) [15]. SMPs are lightweight, low-priced, easily tunable, and more deformable compared to shape memory alloys (SMAs) [8,10,16]. A wide range of mechanical and thermal properties such as elastic moduli, yield strengths, and transition temperatures can be imparted by polymers [8]. SMPs exhibit higher compatibility with biological tissues, thereby reducing mechanical mismatches, because of their better mechanical properties (softness and flexibility) than the typically stiff SMAs [9]. The high strain deformability of polymers increases the capable range of shape recovery [8,10], resulting in a further reduction in device size, and the shape recovery temperature of polymers can be easily adjusted to the desired level because of their easier synthesis and lower melting temperature than metal alloys [10]. The biodegradability and biocompatibility of some groups of polymers make SMPs advantageous over SMAs [6,8,9,10].

Biodegradable polymers such as poly(lactic acid) (PLA), poly(lactide-co-glycolide) (PLGA), poly(caprolactone) (PCL), and polyurethane (PU) and their copolymers are promising materials for biomedical applications because of their biodegradability, biocompatibility, and low toxicity of byproducts. The biodegradability of the polymers addresses the concerns arising from the long-term presence of implanted biomedical devices. For example, metallic stents without biodegradability are prone to restenosis and renarrowing of blood vessels if they reside in the body for more than six months [17], and the risk of cytotoxicity due to corrosion products, depending on the exposure period in corrosive media and concentration of byproducts, was reported [7,18]. The need for secondary surgery to remove implanted biodegradable-polymer-based devices after the completion of medical treatments is eliminated because of their biodegradability. In addition, biodegradability may play a functional role in some applications such as drug-eluting systems [17]. SMPs produced using biodegradable polymers have been reported in various biomedical applications, verifying their applicability toward in vitro and in vivo experiments [19,20].

In this review, the basic principle of SMPs is briefly introduced, and the design strategies required to induce the thermomechanical properties and shape memory performance of biodegradable shape memory polymers (BSMPs) are discussed. Major degradation mechanisms and kinetics of representative biodegradable polymers used for BSMPs in the body are introduced. Changes in the properties and performances of BSMPs during the degradation process are discussed. This paper reviews biomedical applications for self-expending stents, drug-eluting systems, and tissue engineering. In particular, many studies aimed at reducing the shape recovery time of stents and combining stents with drug release ability are reviewed.

## 2. Control of properties of BSMPs

BSMPs with various properties have been designed for use in extended biomedical applications, such as suture threads [21,22,23], drug delivery vehicles [24,25,26,27], stents [28,29,30,31,32], and tissue engineering [33,34,35]. The most common principle to program the SME involves the utilization of a polymer network consisting of hard (net points) and soft (switchable) segments [36,37,38,39]. The programing process of the SME involves shape deforming, fixing and eliminating external forces. Soft segments respond to external stimuli such as temperature [24,29,40,41,42], water [43,44], ultrasound [25,45] and others [31,46,47], forming additional cross links or undergoing crystallization to affix a temporary shape in the fixing stage. The hard segments are permanent networks and remain unchanged during the programing process [48,49,50]. This region generates a spring that stores elastic potential energy, which is the driving force for recovering the original shape, when the soft segments are triggered by external stimuli [48,49,50].

Figure 1 shows a representative mechanism for temperature-controlled SMPs. The mobility of soft segments with relatively low glass transition temperatures (T_g_) or melting temperatures (Tm) significantly increases above the transition temperature (T_t_) [36,49,51]. Cooling of the soft segments whose shape is deformed by stress enables the formation of temporary crosslinks or crystallization, maintaining the deformed shape [36,49,50,51]. The permanent polymer network comprises hard segments with high T_t_ [36,49,50,51]. This network, unaffected at T_t_, stores the elastic potential energy for recovering its original shape [8,9,36]. When the temperature exceeds T_t_ again, the energy is released, recovering the original shape as the mobility of soft segments increases [8,9,36].

Shape memory performance is typically evaluated by calculating the shape fixity strain (*R_f_*) and shape recovery strain (*R_r_*). The parameters are defined in Equations (1) and (2) below: shape memory programing proceeds with the applied strain ($\varepsilon_{l}$) by loading, the remaining strain ($\varepsilon_{u}$) as the deformed shape after unloading, and the recovered strain ($\varepsilon_{r}$) over T_t_ [51]. The shape fixity strain represents the extent to which the temporary shape is fixed as programed, which is undertaken by the soft segment [51,52]. The shape recovery strain indicates the extent to which the material recovers its original shape from the temporary shape, which is the role of the hard segment [51,52].
(1)$$R_{f}=\frac{\varepsilon_{u}}{\varepsilon_{l}},$$
(2)$$R_{r}=\frac{\varepsilon_{u}-\varepsilon_{r}}{\varepsilon_{u}},$$

The combination and ratio of the hard and soft segments in BSMP vary with the T_t_, shape memory performance, mechanical properties, and degradation rate. Table 1 lists some BSMPs with several combinations of segments and their properties.

Achieving a T_t_ close to physiological temperature is the primary goal for biomedical SMPs triggered by temperature, and discovering an appropriate combination of soft and hard segments is a generally utilized strategy. For example, PLA, with a T_g_ ranging from 40 to 60 °C, is an attractive material for bioapplication that shows both biocompatibility and biodegradability [8,62]. T_g_ needs to be lowered to achieve the shape memory characteristics, which are triggered at the body temperature. Polypropylene glycol (PPG) (T_g_
=−73 ℃) can be added to lower the T_t_ to room temperature [58], providing soft segments and lowering the T_g_ of the polymer network. Similarly, Zheng et al. [53] proposed biodegradable self-expandable stents fabricated with a polymer blend of poly(propylene carbonate) (PPC) and PCL. In this study, PPC acts as a soft segment because its T_g_ fluctuates between the room and physiological temperatures and is lower than the potential T_t_ of PLC (59–64 °C); thus, it can be utilized for shape recovery. The crystal phase of PCL became a hard segment and controlled the recovery of the PPC phase by limiting mobility. They showed increasing the PCL content increased the T_g_ of PCC/PCL up to 37 °C as shown in Figure 2a, while the Tm remains practically constant. A gradual increase in R_f_ with an increasing PPC content supported the PPC segment in fixing the temporary shape [53].

The degree of crystallinity of semicrystalline soft segments directly influences T_t_ and is achieved by altering the molecular weight of the soft segments, that is, by changing the soft-to-hard segment ratio. A report by Han et al. [25] shows the relationship between molecular weight, crystallinity, and T_t_. They prepared polyurethane block copolymers synthesized by combining 4, 4′-diphenylmethane diisocyanate (MDI) and 1, 4-butanediol (BDO) with PCL. In their study, a controlled increase in the number averaged molecular weight (Mn) of the PCL (soft) segment from 3000 to 8000 g·mol^−1^ increased the crystallinity from 13.4% to 25.4% and elevated the Tm of PCL from 39.6 to 52.3 °C. Le et al. similarly showed the effect of altering the Mw by manipulating the relative amount of the monomer and initiator [63]. They showed the high Mw of PCL resulted in a long polymer chain, leading to a high Tm due to the formation of stabilized crystallites. A polymer network with a Mw of 12,500 g·mol^−1^ exhibited a T_t_ of 36 °C and excellent shape memory performance with R_f_ and R_r_ of 99% and 98%, respectively in the study. The addition of a nucleating agent to aid crystallization can be used to control T_t_ [25]. For example, adding 4 wt% of copper sulfate particles to assist in the crystallization of PCL increases the Tm by 6 °C, compared to that of a polymer of the same molecular weight synthesized without a nucleating agent [25].

Combining comonomers significantly changes the mechanical properties and shape memory performance of the polymer, as summarized in Figure 2b–d representative sample of thermoplastic polyurethane (TPU)/PCL blends studied by Jing [40] showed the effect of the content ratio of the hard and soft segments on the shape memory performance. In this study, increasing the PCL phase improved the mechanical properties and R_f_; however, it yields a low R_r_. The changes in the shape memory performance showed that the crystalline region in PCL acts as a switch segment, thereby fixing the temporary shape, whereas the rubber-like PU segments store the energy for recovering the original shape. The melting temperature increased as the PCL ratio increases, owing to the high crystallinity of the PCL phase [40]. AB-copolymers made of PLGA with different acrylates functioning as comonomers exhibit improved thermal and mechanical properties [42]. Methacryloyl chloride groups interacting with ethyl acrylate (EtA), butyl acrylate (BuA), or hexyl acrylate act as hard segments, when PLGA was the soft segment. The addition of acrylate enhances the elastic properties and toughness of the polymer network over a homonetwork of poly [(L-lactide)-ran-glycolide] dimethacrylates, which are brittle below a T_g_ of 55 °C [42]. Additionally, the mechanical properties of lactide-based polymers generally tend to be significantly decreased by hydration because of plasticization by water [64]. This challenge in conventional degradable polymers was overcome by introducing polyethylene glycol (PEG) crystallization [57]: the hydrophilic nature of the PEG domain stiffens the polymer network, resulting in an increase in the elastic modulus of the water. The increasing elasticity and degradation rate were controlled using the molar ratio between lactic acid and glycolic acid. PEG, which acts as a soft segment, lowered T_t_ to the body temperature [57].

Controlling the chain length and cross-linking density was studied as a strategy to improving the tensile properties. Yang et al. designed isosorbide (ISO)-PUs based SMPs with different chain length of diisocyanates from ISO and hexamethylene diisocyanate (HDI), which results in different hard segment contents [66]. The increase in hard-segment length by synthesizing diisocyanate with different repeated numbers of ISO molecule enhanced the elastic modulus, tensile and compression strength of SMPs at body temperature. The tensile properties of PUs from three ISO molecules doubled over those of the polymer from an ISO molecule, with little drop in elongation at break [66]. Seo et al. proposed soybean oil and polycaprolactone (SC)-based SMPs with enhanced mechanical properties by introducing polyrotaxane (PRX) as cross-linker [67]. The methacrylate group in PRX increased crosslinking density and flexibility, tripling and even quadrupling both tensile strength and strain simultaneously over little change in elastic modulus compared to SC without PRX. The improvements in mechanical properties and flexibility of SMPs are due to hydroxyl groups in PRX forming large amount of cross-linking points, and those can be made to slide by applied stress [67].

## 3. Biodegradable Behaviors of BSMPs

The biodegradability of each component determines the dissolution characteristics of the BSMPs. The biodegradation mechanisms of polyesters, PUs, and polyanhydrides are reviewed as representative biodegradable polymers, which have been widely utilized and investigated as SMPs. The mechanisms, products, and rates of degradation vary with the functional groups in the polymer, reactive with water and oxygen [68]. Hydrolysis and oxidation are the major degradation mechanisms in biomedical applications. The polymer erodes through two different ways, surface and bulk erosion, depending on the water diffusivity in the polymer matrix and the degradation rate of the polymer [69,70].

Polyester-based polymers have been extensively used and investigated among biodegradable polymers [62,71,72,73,74,75]. This group of polymers are generally degraded by hydrolysis in biofluids because of their chemical structure [62,71,72,75]. First, the ester bond undergoes a hydrolytic cleavage through the penetration of aqueous fluid into the polymer [62,71,72]. The cleaved ester bonds are then converted to carboxyl end groups, as shown in Figure 3a, thus accelerating the hydrolytic reaction with other ester bonds [76]. The autocatalysis of carboxyl end groups generally occurs during the hydrolysis of polymers with ester bonds [68]. As water uptake is critical in this reaction, the degradation process dominates in amorphous domains, where permeation is easier than in crystalline polymer regions [71].

The degradation of ester bonds proceeds with slight weight loss for some time after the initial stage [68]. Only a few degraded oligomers near the surface are mobile because the polymer chains are not scissored sufficiently enough to allow the degraded oligomer in the interior matrix to escape [69]. The degraded parts start to diffuse and dissolve into fluids when the molecular weight decreases sufficiently, and an overall weight loss begins [78]. Ester bonds located away from the surface are degraded faster than those near the surface. This occurs because the density of the carboxyl end groups is higher in the interior than on the exterior of the polymer networks [79]. This type of erosion is called bulk erosion, as shown in Figure 4b [80]. Therefore, the degradation rate is dependent on the physical properties of the device, in particular, the thickness and surface-to-volume ratio [68,79]. Consequently, the degradation of solid PLA is virtually independent of the pH level in the human body because $H^{+}$ and ${\mathrm{OH}}^{-}$ ions diffuse with difficulty into the bulk [68]. Semicrystalline polymers with ester bonds, such as PLGA (Figure 3b) and PCL, show a similar degradation behavior, generating carboxyl end groups from ester bonds as PLA.

Crystallinity is a key factor in refining the degradation properties of semicrystalline polymers [55,56,59]. For example, the microstructural phase ratio was controlled by the PLA stereoisomer content to modify the degradation and mechanical properties because semicrystalline poly (l, l-lactide) (PLLA) and poly (d-lactide) (PDLA) exhibit lower degradation rates than amorphous poly (dl-lactic acid) (PDLLA) [8]. A PDLLA solution blended with other stereoisomers enhances the shape memory performance and decreases the enzymatic degradation rate with the addition of the semicrystalline content [59]. Another PLA-stereoisomer-based polymer elastomer exhibits an increased thermal degradation temperature with an increased PLLA to PDLLA ratio [55,56]. Reducing the chemical crosslinking density and restricting the mobility of the PLA chain can enhance the degradation speed [81]. In addition, using hydrophilic or hydrophobic copolymers can improve degradation properties. For example, PCL generally exhibits a lower degradation rate than PLGA and PLA because of its hydrophobicity and high crystallinity. The combination of PCL with hydrophilic copolymers allows the acceleration of degradation [55].

Polyanhydrides show rapid hydrolytic degradation by the most reactive functional groups in an aqueous environment [70]. Anhydride groups react with water molecules to generate products with carboxyl end groups, similar to ester hydrolysis [70,82,83]. However, polyanhydrides were observed to be surface eroding, undergoing heterogeneous degradation from a near surface, as shown in Figure 4a, because the water uptake process in the polymer is slower than hydrolytic degradation [70]. The erosion type is switched from surface to bulk erosion when the thickness of the device is reduced to a certain depth [68,69,70]. The transition thickness estimated by comparing the diffusion and degradation rates of water supports the occurrence of different erosions in polymers: surface erosion dominates polyanhydrides (20 μm) and bulk erosion dominates PLA (40 mm) [68]. Surface erosion provides a linear degradation rate; thus, polymers with these properties can be utilized as pacemakers to control degradation speed [26]. Xiao et al. introduced poly (sebacic anhydride) to accelerate the degradation rate of PCL [26]. This study confirmed that drug release and degradation rates were enhanced with slight reduction in shape memory performance.

PUs, another polymer group widely used for SMPs, are less reactive than the polymers with aliphatic ester groups during hydrolytic degradation [68]. Large conjugate structures reduce the hydrolytic activity of carbon by a steric effect [68]. Most PUs generally show good hydrolytic stability, except for those made with polyester diols [77]. The polymer degrades in the same way as PLA: water reacts with ester links to produce carboxyl ends, leading to repetitive reactions, as shown in Figure 3c [77]. Oxidation degrades PUs [poly (ester-urethanes) (PEtUS) and polycarbonate-urethanes (PCUs)] exposed to oxygen molecules in a physiological environment (Figure 5) [77]. In addition, polyether and polyamines contain free radicals that are critical for oxidation [68,77]. Oxygen molecules react with free radicals in the polymer chains to produce two new free radicals [68,77]. These radicals are transferred to other polymer chains, and this process increases the mobility of free radicals [68,77]. The chain with a free radical cleaves onto one chain with a double bond and the other with a free radical [68]. The rate of degradation by oxidation justifiably depends on the concentration of oxygen molecules and free radicals [68,77]. Swelling by water absorption and fatigue by repetitive mechanical loading generally play a role in easing degradation [77]. Water uptake decreases the T_g_ of polymers acting as plasticizers, and the resulting polymer network undergoes a loss in rigidity, elastic modulus, creep resistance [68] and crack growth resistance [84]. Repetitive or chronic loading on polymer networks creates mechanical friction and weakens the linkages between the linked chains [68].

Adequate degradability, taking into account the time scale for shape recovery and maintenance of mechanical properties, needs to be determined. The shape-recovery performance decreases according to the degradation of the polymer network [54,72]. Degradation decreases molecular weight ($M_{w}$) and changes the amorphous-to-crystalline phase ratio, which is related to the ratio of the hard and soft segments. For example, the hydrolysis of PCL blended with styrene-butadienestyrene (SBS) degrades the amorphous phase and decreases the molecular weight of PCL which acts as the hard segment; it increases the crystallinity elevates the T_g_ and Tm of the PCL phase [54]. This study shows that the decrease of molecular weight of hard segments by degradation may decrease the shape recovery rate, as seen in Figure 2e [54]. Samples Mw2 through Mw5 underwent enzymatic hydrolysis for 7, 15, 21 and 25 days respectively; Mw1 is an as-prepared sample [54]. Smaller molecular weight creates shorter PCL chains, increases chain mobility, and decreases the degree of entanglement, which fixes the permanent structure as a hard segment [54]. Pretsch et al. showed the drawbacks in shape recovery rate with cyclic programming process of poly(ester urethane) based SMP as it underwent hydrolytic degradation [72]. In this study, the shape recovery ability of the hydrolyzed polymer deteriorated by 20% compared to the polymer without hydrolysis. An increase in crystallinity of soft segments was observed with degradation, resulting in the increase of $T_{m}$ of soft segments by 15 ℃, and similarly for $T_{t}$ [72].

Mechanical properties also alter as degradation proceeds. Elastic modulus and tensile strength generally decrease if polymer chains shorten and molecular weight drops [65,72]. The predictable effects of degradation on mechanical properties are significant in designing BSMP to maintain proper functionality. Pretsch et al. showed that tensile properties of poly(ester urethane)-based SMP, including elongation at break, tensile strength, and toughness, show serious reduction with hydrolytic degradation. In this research, minor reductions in mechanical properties by water immersion occurred in the first two days [72]. Hydrolytic degradation rapidly worsened the mechanical properties over several more days, with decreases in cross-linking density by chain scission of ester bonds, and then embrittlement was followed by erosion [72]. Serrano et al. showed the effect of hydrolytic degradation on tensile properties of polydiolcitrate-based shape-memory elastomers with different mole ratios of hydroxyl to carboxyl groups [65]. In this study, polymers with the same ratio of functional groups show deterioration in tensile strength and increase in elastic modulus with degradation. Small increases in the hydroxyl ratio and use of a hydrophobic diol enhanced the elastic modulus and tensile strength of the polymer with degradation time, though the elongation at break decreased (Figure 2f) [65].

## 4. Applications of BSMPs

BSMPs exhibit two main properties, namely, shape memory and biodegradability. SME allows minimally invasive surgery and automatic actuation, exploiting stimuli-responsive self-deformability to memorize shapes [24,25,26,27,28,29,30,31,32]. This effect makes the treatment focused on the target region possible with a minimal influence on other organs along the migration routes, which is significant for biomedical treatments, such as drug delivery or stent implantation [24,25,26,27,28,29,30,31,32]. Biodegradable polymers, which naturally decompose in a body after a certain period as designed, can obviate the need for post-surgery, which is necessary for removing implanted devices, after the treatment is complete [85]. Representative biomedical applications are reviewed in this section.

Lendlein et al. in 2002 demonstrated the feasibility of thermo-responsive BSMP suture to close an incision as the photo series in Figure 6a [22]. The smart surgical suture is capable of wound close with proper stress preventing necrosis or scar tissue which is probable when the stress is too strong or weak [22]. A biodegradable polymer based stent with SME has eliminated secondary removal surgery and the risk of restenosis [17], in addition to the self-expanding property that Nitinol facilitates by phase transformation [86], which replaced the conventional balloon-expandable stainless steels stents. Studies to accelerate the shape recovery rate to avoid migration from the targeted location are in progress [31,32,87,88,89]. Tamai et al. investigated the feasibility and safety of PLLA-based stents implanted in patients [87]. The heat for shape recovery was transferred by a dye heated to 80 °C through a stent delivery balloon with an optimal recovery time of 30 s [87]. It was observed through the use of bilayered stents comprising PLLA and PLGA that the T_g_ and thickness of the outer polymer layer affect the shape recovery rate of the stents [88]. Adding a PLGA layer reduced the expansion time from 15 days to a few minutes for pure PLLA [88]. A Tm-based self-expandable stent using a hyperbranched PCL as a switching segment with polyester, shortened the expansion time to 25 s, unlike certain previously reported T_g_-based stents [32]. Stents made from polymers with nanoparticles enhanced the shape recovery performance of PU and reduced the complete recovery time [31,89]. The nanoparticle, Fe_3_O_4_, on PU, transforming electromagnetic energy for heating, enables remote activation, using an alternating magnetic field [31]. Wei et al. demonstrated 4D printed PLA/Fe_3_O_4_ SMPs remote actuation by magnetic field and schematics of potential application as the intravascular stent shown in Figure 6b [90].

The structural integrity of biodegradable polymer stents is another actively discussed topic, considering that the primary goal of stent implantation is to achieve structural maintenance under external pressure, induced by vessel walls [30,31,92,93,94,95]. Laser-etched polyurethane-based stents in several geometries showed collapse pressures comparable to those of commercialized steel stents [92]. Another PU-based stent that adopted a solution-mixing fabrication method provided an increased elastic modulus, radial stiffness, and force recovery [30]. Mechanical performance and structural stability have been studied using numerical methods [94,95]. Stents fabricated by 3D printing are an attractive future solution for patient-personalized designs that optimize mechanical behavior in combination with finite element analysis (FEA) studies [31,93].

BSMPs for drug delivery have been studied using SME or biodegradability to control drug release rates. Loaded drugs can be released by a conduit that opens as the programed shape is recovered [44,45,91] as shown in Figure 6d, or by removing a coated surface by degradation [17] in the physiological environment. Many studies have been conducted to control drug release rates by enhancing drug solubility in water [17], providing multiple intermediate shapes [45], changing drug-loading concentrations [96] or molecular weights [97] and introducing nanoparticles [24,31,89,96]. Incorporating the drug-eluting function with a stent has been proposed as a highly desirable way to prevent restenosis and inflammation. The influence of drug containment on mechanical properties [91], shape memory performance [24], and biodegradation rate [91,98] has been investigated. The breaking stress and strain of a stent prepared by Jaworska et al. containing 7 wt% of a drug decreased to approximately 70% of the stent without the drug because the aggregated drug particles induced notch effects; this study showed that drug incorporation can change stent hydrophilicity and thus, the hydrolysis degradation rate [91,98]. Kashif et al. examined the change in shape memory performance with drug content for stents made of PCL/trisilanolphenyl polyhedral oligomeric silsequioxane (TSP-POSS) nanocomposite films [24]. Shape memory performance decreased (R_f_: 85–81 and R_r_: 94–85) with an increasing drug content from 0 to 10 wt%. Yang et al. fabricated a dual drug-eluting stent using two different drug-loading methods, one by chemical crosslinking conjugation and the other by physical absorption through the formation of a gelation layer [17]. The physically absorbed drug was released within 14 days, whereas the chemically conjugated drug was released sustainably over 60 days [17].

SMP-based tissue engineering applications have been reported with various targets, encouraging the repair and regeneration of patients’ tissues, which will substitute the implanted materials [9,33,34,35]. Hiebl et al. implanted a multiblock copolymer comprising poly (p-dioxanone) and PCL to induce blood vessel formation [19]. The properties of poly (1,10-decanediol-co-citrate) (PDC), which encourage blood vessel formation and produce small degradation particles, are speculated to provide good tissue integration ability [19].

Polymers with enhanced mechanical properties are facilitating the substitution of metals in hard-tissue applications such as bone and tendon repair, with the potential to resolve limitations of metal implantation such as inflammatory response and insufficient biocompatibility [99]. Many strategies to enhance the mechanical properties for polyurethane-based bone tissue engineering have been reported [100,101,102,103]. PU with enhanced tensile and compressive strength approximates bone and tendon tissues by increasing the crosslinks by changing the ratio of monomers or longer UV exposure [101]. Poly-ether-urethane foam with porosity over 70% showed good mechanical stability under cyclic compression, mimicking trabecular bone structure [102]. Introduction of nanocomposites as bone tissue scaffolds has been widely reported to enhance mechanical properties [20,33,60,73,81,104]. Zhang et al. developed a self-fitting SMP scaffold for bone-defect treatment using PCL, including fused salt particles, as shown in Figure 6c [60]. Wang et al. presented a water-responsive shape-memory bone scaffold using a 3D-printed PU containing supermagnetic iron oxide nanoparticles [104]. The nanoparticles encourage osteogenic induction and enhance shape fixity. Zhai et al. introduced a nanoclay into poly (N-acryloyl glycinamide) to enhance osteogenic differentiation [20]. Y. Zhang et al. modified polyurethane-based SMPs with hydroxyapatite (HA) nanofiller, an inorganic component of natural bone tissue, for enhanced tensile properties [99].

## 5. Conclusions

BSMPs contribute to minimally invasive surgery and eliminate the inconvenience of secondary surgery, resulting in a reduced risk for excessive bleeding and infection. Additionally, their shape deformation and degradation in the body are excellent and can be used as an actuating strategy in biomedical devices. A wide range of thermomechanical properties and degradation rates can be achieved by combining different biodegradable polymers. Complete and rapid shape recovery can be achieved by adjusting crystallinity and adding nanoparticles. The mechanical properties and shape-memory performance show time-dependent changes because of accelerated degradation in the form of surface or bulk erosion. BSMPs are appropriately designed by considering the required properties and performance and are utilized for various biomedical devices such as stents, drug-delivery, and tissue-engineering applications. BSMPs have the potential to be used to build electronic systems with functions that can be tuned according to shape transitions. The performance and applicability of BSMPs may be expanded by integration with transient electronic devices and soft robotics, and this is expected to provide new opportunities for ecofriendly probiotic robots and device technologies. The expansion of biodegradable-polymer-based SMPs may lead to ecofriendly soft-robotic-based technologies that minimize robotic waste.

## Figures and Tables

**Figure 1 micromachines-12-00757-f001:**
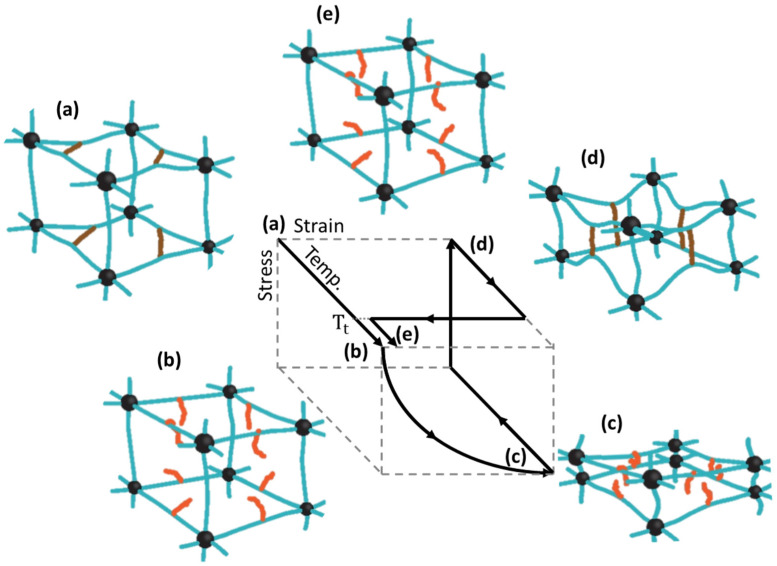
Schematic representation of programming process for thermo-responsive shape memory polymer with schematic of thermomechanical cyclic. Black dots and blue lines: hard segment; brown lines: soft segment below T_t_; orange lines: soft segment above T_t_. (**a**) Initial shape; (**b**) shape at the elevated temperature (**c**) shape mechanically deformed at the elevated temperature; (**d**) shape fixed as deformed by cooling; and (**e**) shape recovered by heating.

**Figure 2 micromachines-12-00757-f002:**
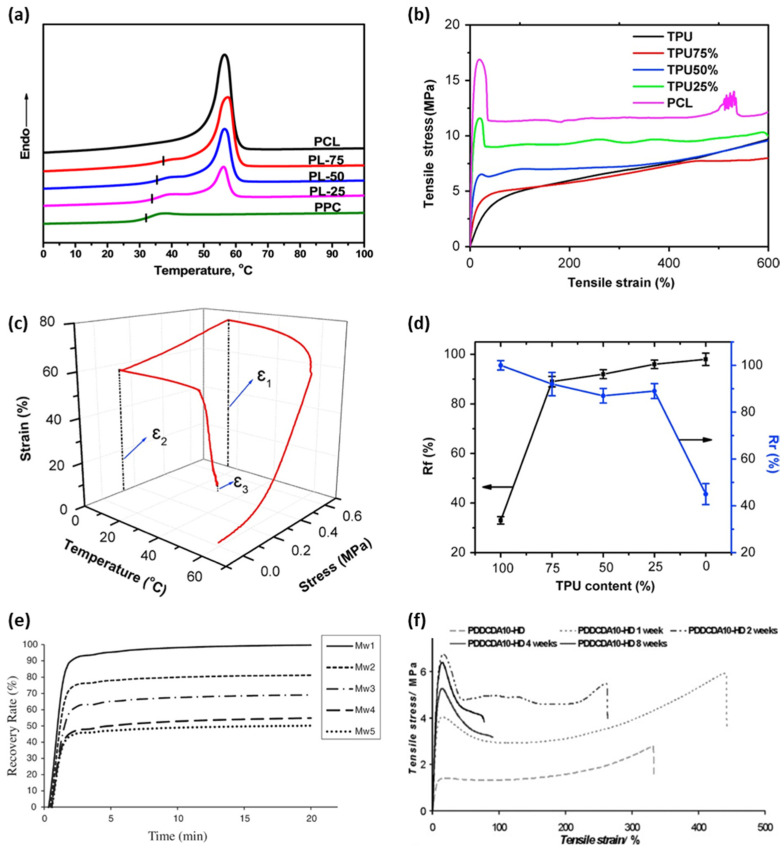
Thermomechanical properties and shape memory performance of BSMPs (**a**) DSC heating curves of PCL, PPC, and the PPC/PCL blends; (adapted with permission [53] ©2017, American Chemical Society) (**b**) stress–strain curves of TPU/PCL blends; (**c**) thermomechanical cyclic schematic of TPU25%; (**d**) shape memory performances of TPU25% (adapted with permission [40] ©2016, Elsevier Ltd.); (**e**) shape recovery rate curves of poly(caprolactone)/styrene-butadiene-styrene blend for different molecular weights by degradation (adapted with permission [54] © 2020, Wiley Periodicals LLC); (**f**) effect of polymer degradation on the mechanical properties of hydroxyl-dominant polymers (adapted with permission [65] ©2011, WILEY-VCH Verlag GmbH & Co. KGaA, Weinheim).

**Figure 3 micromachines-12-00757-f003:**
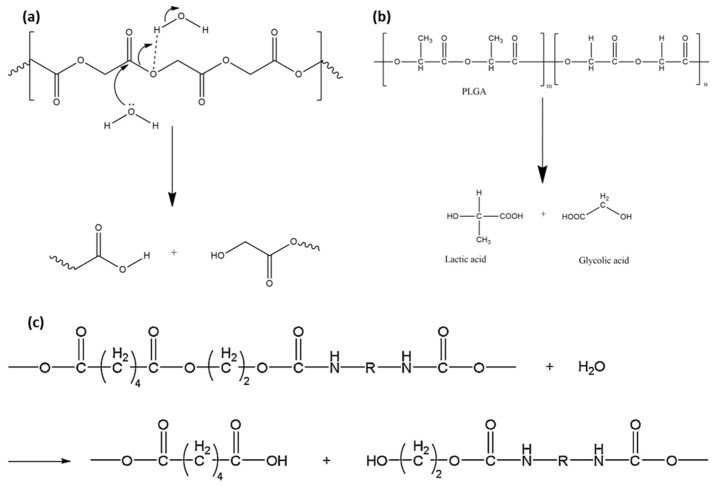
Degradation of ester bond based polymers. (**a**) Hydrolytic cleavage of ester bonds; (**b**) degradation of PLGA (adapted with permission [76] ©2017, Elsevier); (**c**) hydrolysis of poly(ester-urethane)s (adapted with permission [77] ©2018, Elsevier B.V.).

**Figure 4 micromachines-12-00757-f004:**
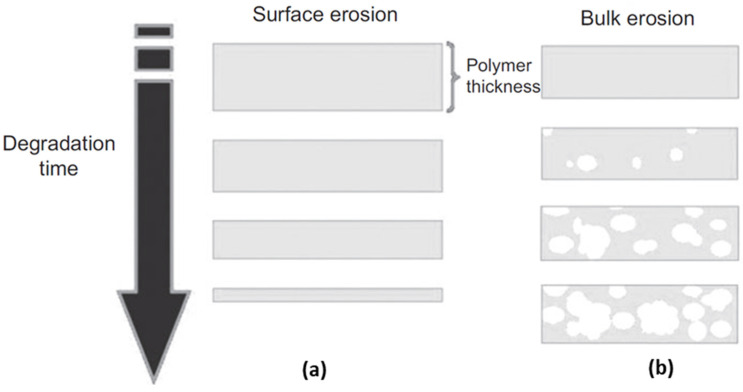
Schematic illustration of two types of erosion: (**a**) surface erosion and (**b**) bulk erosion with autocatalysis (adapted with permission [80] ©2013, Walter de Gruyter Berlin Boston).

**Figure 5 micromachines-12-00757-f005:**
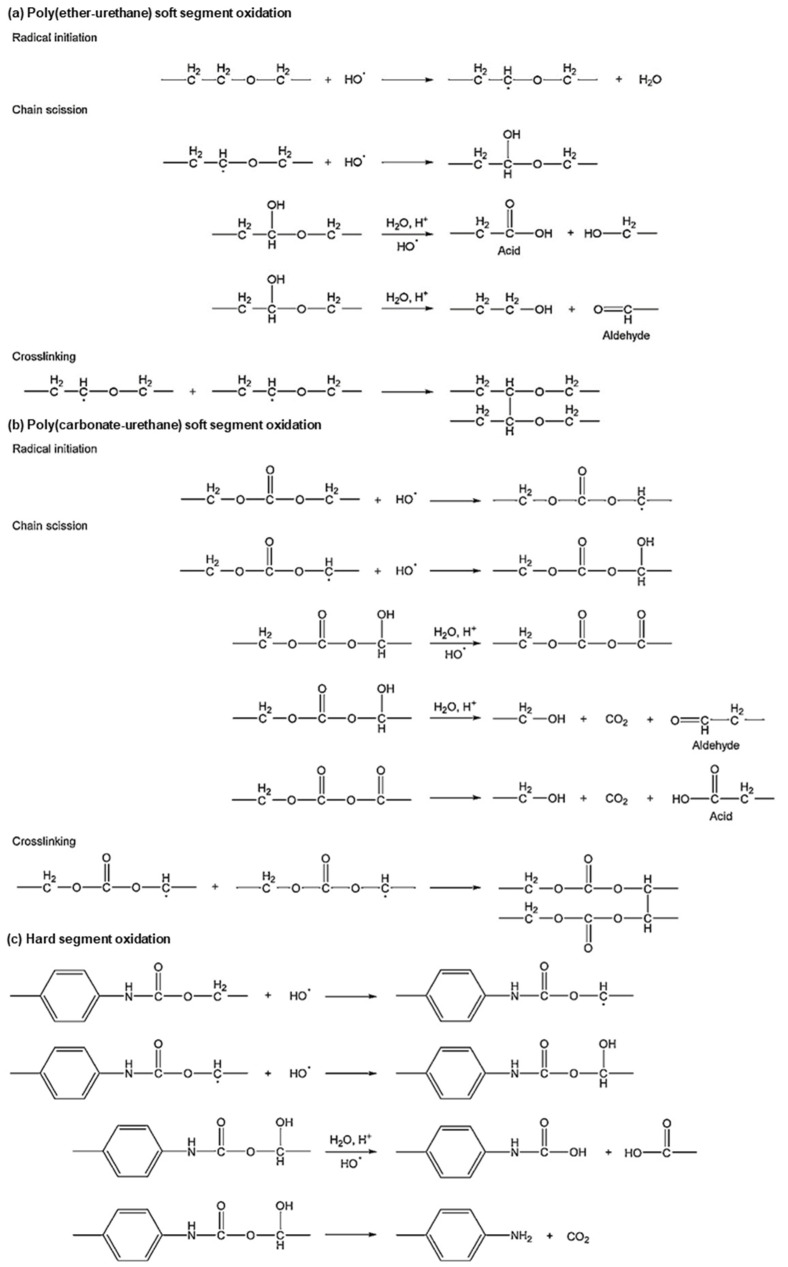
Oxidation of poly(ether-urethane)s and poly(carbonate-urethane)s (adapted with permission [77] ©2018, Elsevier B.V.).

**Figure 6 micromachines-12-00757-f006:**
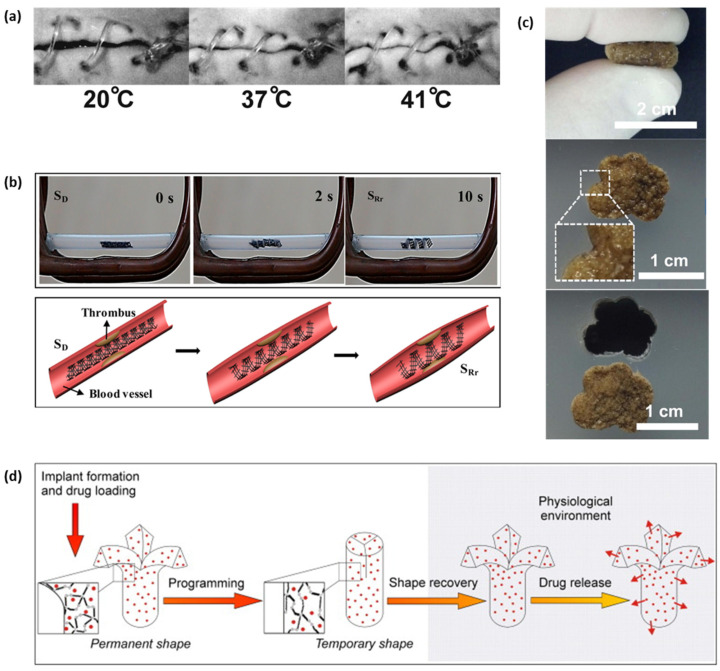
Biomedical applications of BSMPs. (**a**) The photo series of suture for wound closure from the animal experiment shows (**left** to **right**) the shrinkage of the fiber while temperature increases. (Adapted with permission [22] © 2002, American Association for the Advancement of Science); (**b**) demonstration of the restrictive shape recovery process triggered by a 30 kHz alternating magnetic field; potential application of the 4D scaffold as an intravascular stent. Here, the deformation temperature was 80 °C. Therefore, S_D_, and S_Rr_ represent for the original, deformed, and recovery shapes under restrictive conditions, respectively (adapted with permission [90] ©2017, American Chemical Society); (**c**) self-fitting behavior of a polydopamine-coated PCL SMP scaffold (adapted and reproduced with permission [60] © 2014, Elsevier); (**d**) concept of programming, shape recovery, and drug release of drug loaded SMP devices (adapted with permission [91] ©2009, Elsevier B.V.).

**Table 1 micromachines-12-00757-t001:** Thermomechanical properties and shape memory performance of BSMPs with different combinations of segments.

Material	Hard Segment	Soft Segment	$T_{\mathrm{trans}}$ (°C)	$R_{f}$	$R_{r}$	$dR_{r}/dt$	E (MPa)	Ref
PPC/PCL	PCL	PPC	37	97	94	6%/s	-	[53]
PCL-SBS	PCL	SBS	58	-	100	1%/min	116	[54]
PCL/PU	PU	PCL	45	-	77	1.3°/s	4.3–9	[30]
PCL-PLLA	PLLA	PCL	85	-	-	-	120–180	[55]
Fe_3_O_4_-PCLAU	PLA	PCL	40	99	82–85	0.2%/s	-	[31]
Fe_3_O_4_-PLA	Crystallites	Amorphous	65–67		90–95	4%/s	2–3 × 10^3^	[47]
PLA	Crystallites	Amorphous	70	99	87–96	-	4 × 10^3^	[29]
PLDU	PLLA	PDLLA	38–46	99	72–95	-	1.5–2 × 10^3^	[56]
PLGA-PEG	PLGA	PEG	37	99	99	3–10%/min	206–330	[57]
PLGA/PEGDA	PLGA	PEG	37	-	82–92	0.7%/s	1.2–1.6 × 10^3^	[43]
PCL-PEG	PCL	PEG	41–45	100	88–100	9–10%/s	28–97	[17]
GelUPy	Gelatin	UPy	100	-	-	-	2.42	[44]
PLA-PPG	PLA	PPG		92–96	87–99.5	2.6–3.5%/K	-	[58]
PDLLA-co-TMC	DDLA	TMC	37–44	99	94–100	-	175–200	[33]
PCL/TspPOSS	OPD	PCL	70	81	85	-	-	[24]
TPU/PCL	TPU	PCL	32	98	90	-	-	[40]
Sc-PLA-PDLLA	Sc-PLA	PDLLA	70		65–99	4.5–5%/s	-	[59]
PCL-DA	DA	PCL	54–60	100	95–100	-	0.54–4.3	[35,60]
PLGA-EA	EA	PLA	20–50	97–98	99–100	-	1.6–288	[42]
PLGA-BA	BA	PLA	−10–40	93–97	84–100	-	3.3–30	[42]
PLGA-HA	HA	PLA	−30–60	91–96	97–99	-	12–37	[42]
PCL-MDI-BDO	MDI-BDO	PCL	36–52	99	99	4%/s	-	[25]
PCL-PHBV	PHVB	PCL	40	94	98	4%/s	42–70	[32]
PCL-PPDO	PPDO	PCL	37	99	97–99	1.5%/s	407–542	[61]
ICM/PCL	ICM	PCL	60	89–100	85–100	-	-	[41]
OCL-HDI	HDI	OCL	37–39	98	99	-	-	[27]
OCL-ODX	ODX	OCL	40	98–99.5	76–99	5%/s	-	[22]

PPC = polypropylene carbonate, PCL = poly(ε-caprolactone), SBS = styrene butadiene styrene, PU = polyurethane, PLLA = poly(l-lactide), PLA = poly(lactide), PLGA = poly(lactide-co-glycolide), PEG = polyethylene glycol, Gel = gelatin, UPy = ureido-pyrimidinone, PPG = polypropylene glycol, TMC = 1,3-trimethylene carbonate, TspPOSS = trisilanol phenyl polyhedral oligomeric silsequioxane, TPU = thermoplastic polyurethane, Sc-PLA = stereocomplex PLA, PDLLA = poly(d,l-lactide), DA = diacrylate, EA = ethyl acrylate, BA = butyl acrylate, HA = hexyl acrylate, MDI = methylene, diphenyl 4,4′-diisocyanate, BDO = 1,4-butanediol, PHBV = poly[(R)-3-hydroxybutyrate-co-(R)-3-hydroxyvalerate], PPDO = poly(p-dioxanone), ICM = 7-(3,5-dicarboxyphenoxy) carbonylmethoxycoumarin, OCL = oligo(ε-caprolactone), HDI = hexamethylene diisocyanate, ODX = oligo(p-dioxanone), OPD = poly(2-oxepane-1,5-dione).

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
