# Peer review of "Principles for Controlling the Shape Recovery and Degradation Behavior of Biodegradable Shape-Memory Polymers in Biomedical Applications"

_micromachines, 2021, doi:10.3390/mi12070757_

Round 1
Reviewer 1 Report
This review is well summarising recent biodegradable shape memopoly polymers, properties, degradation mechanism, and application. These contents may be useful for readers. I recommned this manuscript will be published on Micromachines.
Please check some typo, such as,
Line 333-334, "... based stent stent with SME ..."
Reviewer 2 Report
Dear Authors
As a review paper its important to cover and concise the content. Although author focus on chemical structure, biological behaviour, drug release, however mechanical part is lacking in explanation. Second at some point in conclusion authors has mention about 3D and 4D printing. So the whole idea is somehow scatter not composed, the title should be more focus in the area of thermal, chemical and biological application. As mechanical part is not well explained.
So herewith i suggest authors should carry out major revision for suitable publication.
Round 2
Reviewer 2 Report
The revised version of the article is significantly improved in terms of content and title. However before it goes to the stage of publication, author should add one paragraph of advantages and benefit of choosing shape memory polymer over shape memory alloys and combination of shape memory alloy with polymer. Please refer some of the articles as below
1. Study of Interfacial Adhesion between Nickel-Titanium Shape Memory Alloy and a Polymer Matrix by Laser Surface Pattern. Appl. Sci. 2020, 10, 2172. https://doi.org/10.3390/app10062172
2. Net-Shape NiTi Shape Memory Alloy by Spark Plasma Sintering Method. Appl. Sci. 2021, 11, 1802. https://doi.org/10.3390/app11041802
